# Alteration of Gut Microbes in Benign Prostatic Hyperplasia Model and Finasteride Treatment Model

**DOI:** 10.3390/ijms24065904

**Published:** 2023-03-21

**Authors:** Jinho An, Youngcheon Song, Sangbum Kim, Hyunseok Kong, Kyungjae Kim

**Affiliations:** 1College of Pharmacy, Sahmyook University, Seoul 01795, Republic of Korea; 2PADAM Natural Material Research Institute, Sahmyook University, Seoul 01795, Republic of Korea; 3College of Animal Biotechnology and Resource, Sahmyook University, Seoul 01795, Republic of Korea

**Keywords:** gut microbes, benign prostatic hyperplasia, apoptosis, finasteride, *Lactobacillus*, *Acetatifactor*

## Abstract

Gut microbes are closely associated with disease onset and improvement. However, the effects of gut microbes on the occurrence, prevention, and treatment of benign prostatic hyperplasia (BPH) are still unclear. We investigated the alteration of gut microbiota with implications for the diagnosis, prevention, and treatment of BPH and identified correlations among various indicators, including hormone indicators, apoptosis markers in BPH, and finasteride treatment models. BPH induction altered the abundance of *Lactobacillus*, *Flavonifractor*, *Acetatifactor*, *Oscillibacter*, *Pseudoflavonifractor*, *Intestinimonas*, and *Butyricimonas* genera, which are related to BPH indicators. Among these, the altered abundance of *Lactobacillus* and *Acetatifactor* was associated with the promotion and inhibition of prostate apoptosis, respectively. Finasteride treatment altered the abundance of *Barnesiella*, *Acetatifactor*, *Butyricimonas*, *Desulfovibrio*, *Anaerobacterium*, and *Robinsoniella* genera, which are related to BPH indicators. Among these, altered abundances of *Desulfovibrio* and *Acetatifactor* were associated with the promotion and inhibition of prostate apoptosis, respectively. In addition, the abundances of *Lactobacillus* and *Acetatifactor* were normalized after finasteride treatment. In conclusion, the association between apoptosis and altered abundances of *Lactobacillus* and *Acetatifactor*, among other gut microbes, suggests their potential utility in the diagnosis, prevention, and treatment of BPH.

## 1. Introduction

The microbiome is an important factor affecting human health. The microbiome is composed of bacteria, archaea, viruses, and eukaryotic microorganisms inside and outside the body. These microbes exert physiological effects on the entire human body, such as participation in metabolic functions for health maintenance and disease treatment, protecting the body from pathogens, and influencing the immune system. Recently, various applications, such as probiotics and prebiotics targeting the microbiome, have been developed for the prevention or treatment of various diseases. In addition, the importance of gut microbes, which are closely related to immunity, has been magnified by the onset of COVID-19 and the subsequent mutant variants [1]. The human microbiome is acquired during the first few years of life, and the majority of the microbiome, similar to that of an adult, is formed as early as the first year of life [2]. However, it is susceptible to change due to the external environment, including diet and drug treatments. These changes can affect health and contribute to susceptibility to diseases such as diabetes, cardiovascular diseases, allergies, inflammatory bowel diseases, and various cancers [3,4].

Benign prostatic hyperplasia (BPH) is a common lower urinary tract symptom in middle-aged men and is closely related to the quality of life in men. An increasing lifespan has resulted in an increased prevalence of BPH worldwide. BPH is reported in approximately 50% of patients aged 50 years or older [5], and, according to another report, the estimated prevalence of BPH is 26.2% of the world population [6]. BPH is closely related to various lower urinary tract symptoms (LUTSs), such as urinary incontinence, urgency, and frequency; acute urinary retention; urinary hemorrhage; urethral infection; bladder stones; and renal failure [7,8]. Testosterone (TE) secretion decreases with aging. TE is reduced to dihydrotestosterone (DHT), which has a 2–3-fold or 15–30-fold greater sensitivity to androgen receptor (AR) than TE or adrenal androgens. It acts as an endogenous ligand with high biological activity in the presence of 5α-reductase [9]. However, as DHT contributes to the formation of gonads during pregnancy, accumulated DHT is a main cause of BPH [10]. BPH is characterized by an increased prostate profile with high 5α-reductase and DHT levels. Additionally, increased levels of the AR and prostate-specific antigen (PSA), antiapoptosis in the prostate, inflammatory response, and chronic inflammation have been reported [11,12,13,14,15].

Most previous studies on the relationship between the microbiome and prostate diseases have focused on direct interactions such as an infection of the prostate [16,17]; however, the association between BPH and the gut microbiota is still unclear. Moreover, despite the therapeutic effect of finasteride, which acts as a type 2 5α-reductase inhibitor as a BPH drug, its effect on the gut microbiome remains unclear. Various side effects of finasteride, such as finasteride syndrome (postfinasteride syndrome), sexual desire, sexual dysfunction, depression, anxiety, and suicidal ideation, have been reported [18]. Accordingly, gut microbes identified to play a role in disease management may be applied to ameliorate BPH without side effects.

To confirm the hypothesis that alteration of the gut microbiota can potentially aid in the diagnosis, prevention, and treatment of BPH, we investigated the composition of the gut microbiota in BPH and finasteride treatment models. The correlations between altered microbes and various indicators were also analyzed to confirm that certain microbes can potentially aid in the diagnosis, prevention, and treatment of BPH.

## 2. Results

### 2.1. Validation of the Animal Model

Various indicators were identified to evaluate the BPH induction and finasteride treatment. In the BPH group, the body weight and the weight of the liver, thymus, and spleen were significantly decreased (Appendix A), the kidney weight was significantly increased (Appendix A), and the creatinine level was significantly decreased (Appendix A). The prostate profiles, which are most important in diagnosing BPH, were significantly increased in the BPH group and significantly decreased by finasteride treatment (Appendix A). In addition, the levels of 5α-reductase and DHT in the serum were significantly increased in the BPH group and significantly decreased by finasteride treatment (Appendix A). The levels of PSA and AR in the prostate tended to be increased in the BPH group. The PSA level tended to be decreased, and the AR level was significantly decreased by finasteride treatment (Appendix A). In addition, in the BPH group, the expression of Bax was significantly decreased, and Bcl-2 was significantly increased. The expressions of caspase9 and caspase3 tended to be decreased, and the inflammatory cytokines tended to be increased (Appendix A). In the finasteride treatment group, the expressions of Bax and caspase9 were significantly increased, and Bcl-2 significantly decreased. The expressions of caspase3 tended to be increased, and the inflammatory cytokines tended to be decreased (Appendix A).

### 2.2. Effect of BPH Induction and Finasteride Treatment on Gut Microbiota Composition

A total of 1,900,022 sequences were generated from 21 samples, and an average of 5574 ± 642 sequences were included for comparative analysis. No significant differences were identified between the groups (Figure 1a). The alpha diversity of the gut microbiota was analyzed using the Shannon index, which represents evenness, to confirm the diversity in each group. No significant differences were identified between the groups (Figure 1b). To confirm the similarity of the microbial diversity between samples, beta diversity was confirmed through multidimensional scaling (MDS) and principal coordinate analysis (PCoA) (Figure 1c,d). In the PCoA plot, the BPH and BPH + Fina groups are clearly separated (Figure 1d).

A heat map was generated to compare the relative abundance of the top 15 families and top 20 genera. As a result of hierarchical clustering, all microbes were within the phyla Bacteroidota, Verrucomicrobiota, Pseudomonadota, and Bacillota, except Ruminococcaceae. The bacterial phyla Bacteroidota and Bacillota were mainly clustered (Figure 1e,f).

### 2.3. Effects of BPH Induction on Gut Microbiota

The relative abundance in each group was confirmed to select microbes with potential as indicators of BPH diagnosis using linear discriminant analysis effect size (LEfSe) based on Kruskal–Wallis and Wilcoxon tests. Six microbes in the control group and eight in the BPH group were confirmed to have a higher relative abundance. Among these, the abundances of Bacilli, *Lactobacillus*, Lactobacillaceae, *Lachnospiracea_incertae_sedis*, Lactobacillales, and Rhodothermaceae were lower in the BPH group than in the control group (Figure 2a). The abundances of Ruminococcaceae, *Flavonifractor*, *Acetatifactor*, *Oscillibacter*, *Pseudoflavonifractor*, *Intestinimonas*, *Butyricimonas*, and *Anaerotruncus* were higher in the BPH group than in the control group (Figure 2a).

The abundance of the identified microbes was analyzed using the LEfSe method. Among the microbes with decreased abundance, the microbes from the *Lactobacillus* genus—belonging to the taxonomic lineage of *Bacilli*, Lactobacillaceae, and *Lactobacillales*—showed consistently decreased abundances in the BPH group (Appendix A). Among the microbes with increased abundance, the microbes from the *Flavonifractor*, *Acetatifactor*, *Oscillibacter*, *Pseudoflavonifractor*, *Butyricimonas*, and *Anaerotruncus* genera had significantly increased abundances in the BPH group (Appendix A).

Correlation analysis was performed to confirm the correlation between bacterial genera, BPH indicators (Appendix A), and apoptosis markers (Appendix A). The abundances of the *Lactobacillus* and *Lachnospiracea_incertae_sedis* genera decreased in the BPH group and showed a negative correlation with almost all BPH indicators (Figure 2b). Additionally, the abundance of *Lactobacillus* was associated with the progression of apoptosis in the prostate (Figure 2c). On the contrary, the abundances of *Flavonifractor*, *Acetatifactor*, *Oscillibacter*, *Pseudoflavonifractor*, and *Butyricimonas* genera increased in the BPH group and showed overall positive correlations with BPH indicators (Figure 3a). The abundances of *Acetatifactor* and *Butyricimonas* genera were associated with the inhibition of apoptosis in the prostate (Figure 2c).

### 2.4. Effects of Finasteride Treatment on Gut Microbiota

The relative abundance in each group was confirmed using LEfSe based on Kruskal–Wallis and Wilcoxon tests to select the microbes with the potential to prevent or treat BPH. Six microbes in the BPH group and ten microbes in the BPH+Fina group showed higher relative abundances than the rest. The abundances of *Bacterodia*, *Bacterodales*, *Bacterodetes*, *Barnesiella*, *Acetatifactor*, and *Marvinbryantia* were lower in the BPH+Fina group than in the BPH group (Figure 3a); the abundances of *Lachnospi-racea_incertae_sedis*, *Peptococaceae1*, *Peptococcus*, *Butyricimonas*, *Desulfovibrio*, *Anaerobacterium*, *Robinsoniella*, *Ruminococcus2*, and *Actinobacteria* were higher in the BPH+Fina group than in the BPH group (Figure 3a).The abundance of the identified microbes was analyzed using LEfSe. Among the microbes with decreased abundance, the *Barnesiella* genus—belonging to the *Bacterodia* and *Bacterodales* lineage—showed consistently decreased abundance in the BPH+Fina group (Appendix A). The abundances of *Bacterodetes* and *Acetatifactor* significantly decreased in the BPH+Fina group (Appendix A). Among the microbes with increased abundance, the *Butyricimonas*, *Desulfovibrio*, *Anaerobacterium*, and *Robinsoniella* genera showed significantly increased abundances (Appendix A).

Correlation analysis was performed to confirm the correlation between the bacterial genera and the levels of BPH indicators or apoptosis markers. The abundance of *Acetatifactor*, which decreased in the BPH+Fina group, showed an overall positive correlation with the BPH indicators (Figure 3b). These results are similar to those of the correlation analysis that compared the control and BPH groups (Figure 2b). Interestingly, the *Butyricimonas* and *Anaerobacterium* genera, which showed increased abundances in the BPH+Fina group, were clustered with *Acetatifactor*, which showed a similar correlation pattern (Figure 3b). On the contrary, the abundance of *Lachnospiracea_incertae_sedis*, *Robinsoniella*, and *Anaerobacterium* genera, which increased in the BPH+Fina group, showed a negative correlation pattern with the prostate index (Figure 3a). The abundances of *Robinsoniella*, *Anaerobacterium*, and *Desulfovibrio* genera were associated with the progression of prostate apoptosis, whereas the abundances of the *Acetatifactor* and *Marvinbryantia* genera were associated with an inhibition of apoptosis (Figure 3b).

### 2.5. Comparison of Gut Microbes after BPH Induction and Finasteride Treatment

The significant microbial changes were compared using LEfSe based on the Kruskal–Wallis and Wilcoxon tests. A total of 3 microbes in the control group, 4 microbes in the BPH group, and 13 microbes in the BPH+Fina group had a higher relative abundance than each other group (Figure 4a).

Among the microbes that were more abundant in the control group (Figure 4b–e), Bdellovibrionales, Bdellovibrionaceae, and *Vampirovibrio* showed significantly decreased abundances in the BPH+Fina group (Figure 4b–e). Among the microbes that were more abundant in the BPH group (Figure 4f–i), *Flavonifractor*, *Acetatifactor*, and *Oscillibacter* showed significantly increased abundances in the BPH group; however, the abundance of *Acetatifactor* was significantly normalized by the finasteride treatment. The altered abundances of the *Flavonifractor* and *Oscillibacter* genera ere similar to that of *Acetatifactor*. Among the microbes that showed higher abundances in the BPH+Fina group (Figure 4j–v), the abundances of Lactobacillales, Lactobacillaceae, and *Lactobacillus* were significantly normalized, and those of *Butyricimonas*, *Anaerobacterium*, *Desulfovibrio*, and *Robinsoniella* were significantly increased in the BPH+Fina group.

## 3. Discussion

Increasing evidence of the effects of disease or drug treatment on alterations in gut microbiota composition and the effects of some microbial applications on the host demonstrates that changes in gut microbiota composition may play an important role in inducing or ameliorating diseases. These microbes have a significant correlation with the diagnosis, prevention, and treatment of various diseases. However, despite the development of various means for utilizing these microbes, the application of gut microbes in the diagnosis, prevention, and treatment of prostate diseases is still unclear despite increasing worldwide interest due to aging. In addition, although finasteride is a 5α-reductase inhibitor with well-known therapeutic effects, its effects on gut microbiota remain unclear. In this study, we investigated the alteration of gut microbiota composition induced by BPH and finasteride treatments and analyzed the correlation between BPH indicators and apoptosis markers to confirm the presence of microbes, which can contribute to the diagnosis, prevention, and treatment of BPH.

*Lactobacillus*, found in dairy products, is one of the most common probiotic genera, with more than 200 species. Some *Lactobacillus* spp. exert beneficial effects on the human body, such as protecting the host from potential pathogenic invasion and treating diarrhea, vaginal infections, and skin disorders such as eczema [19,20]. In a recent study on the urinary flora of patients with prostate cancer and BPH, a decrease in the abundance of *Lactobacillus iners* or *L. helverticusd* was confirmed, suggesting that these microbes may be biomarkers for the prediction and early treatment of prostate disease [21]. In addition, *Lactobacillus* abundance was decreased in the semen of patients with chronic prostatitis [22]. Another study reported that the intake of fermented milk containing *L. casei* increased the activity of natural killer cells [23], and *Lactobacillus* inhibited human prostate cancer cell tumor necrosis factor related apoptosis-inducing ligand (TRAIL) [24]. However, another study reported conflicting findings that the abundance of *Lactobacillus* in the urine was increased in patients with BPH and prostate cancer [25]. In this study, the induction of BPH using testosterone undecanoate significantly reduced the abundance of *Lactobacillus* compared with that in the normal control group and consistently decreased the abundances of Bacilli, Lactobacillaceae, and Lactobacillales, to which *Lactobacillus* belongs. In addition, the abundance of *Lactobacillus* was significantly increased in the finasteride-treated group and was negatively correlated with prostate profiles, which are the main characteristics of BPH, and the levels of DHT and 5α-reductase. In addition, the abundance of *Lactobacillus* is related to proapoptosis in the prostate tissue. Therefore, despite the lack of evidence, the altered abundance of *Lactobacillus*, in addition to being used for the diagnosis of BPH, may help regulate gut microbes through dietary control, unlike the urinary microbe concentration, which has limitations such as artificial control. This suggests that alterations in the abundance of *Lactobacillus* have the potential to prevent or treat BPH and confirm the therapeutic effect through the regulation of DHT and 5α-reductase and mediation of apoptosis in the prostate tissue.

*Flavonifractor* was reported to increase in patients with radiation enteropathy and is associated with eotaxin [26]. Another study reported that the oral administration of *F. plautii* promoted recovery from acute colitis by inhibiting IL-17 [27]. However, its relationship with BPH requires further study. *Acetatifactor* is a bile-acid-inducing anaerobe that reportedly improves liver function and insulin and glucose tolerance by activating TGR5, which is a bile acid membrane receptor that stimulates glucagon-like peptide-1 secretion [28]. However, other studies have reported that the transplantation of fecal microbiota rich in *Acetatifactor* spp. induced colon inflammation [29] and that *Acetatifactor* mediates liver inflammation, causing abnormalities in lipid metabolism, such as altering the triglyceride, linoleic acid, and glycerophospholipid levels and energy supply processes [29,30]. *Oscillibacter* was reported to be abundant in hosts with impaired renal function, which is a common pathological feature of lower urinary tract ailments, including BPH, and is associated with uremic toxins [31]. *Pseudoflavonifractors* were reported as a type of microbe that significantly contributed to changes in metabolites in obesity-induced type 2 diabetic mice. They have been reported to aggravate metabolic disorders in diabetic patients in relation to their energy metabolism and insulin sensitivity [32]. *Butyricimonas* has been reported to improve hyperglycemia through the regulation of GLP-1R in mice fed a high-fat diet and to be associated with lipid metabolism [33,34,35]. The increased abundance of *Anaerotruncus* is directly related to an increase in inflammatory cytokines due to aging [36]. Our results showed increased abundances of the *Flavonifractor*, *Acetatifactor*, *Oscillibacter*, *Pseudoflavonifractor*, *Butyricimonas*, and *Anaerotruncus* genera in the BPH model, with positive correlations between BPH indicators. Among these microbes, the abundances of *Acetatifactor* and *Butyricimonas* genera are related to the inhibition of apoptosis in the prostate. Among the microbes with an increased abundance following BPH induction, the abundances of *Flavonifractor*, *Acetatifactor*, and *Oscillibacter* genera were normalized through finasteride treatment, compared with that in the BPH model. These results indicate that the altered abundances of *Butyricimonas*, *Acetatifactor*, *Flavonifractor*, *Oscillibacter*, *Pseudoflavonifractor*, and *Anaerotruncus* genera are potential indicators of BPH onset or for evaluating the post-therapy status of BPH.

Additionally, finasteride treatment remarkably increased the abundances of *Butyricimonas*, *Anaerobacterium*, and *Desulfovibrio* genera compared with those in the other groups. Interestingly, although a positive correlation with BPH indicators was confirmed, *Butyricimonas* abundance was further increased by finasteride treatment. Despite recent reports on the positive effects of *Butyricimonas* [33], the abundance of *Butyricimonas* was consistently positively correlated with BPH markers in our study. Therefore, further studies are required. *Anaerobacterium* abundance was reported to be significantly increased by treatment with prednisone, which is a synthetic glucocorticoid widely used for immune-mediated diseases due to its immunosuppressive and anti-inflammatory properties [37]. Because our study did not find significant changes in the expression of inflammatory cytokines, the effect of *Anaerobacterium* on BPH with inflammation requires further investigation. However, the association between the abundance of *Anaerobacterium* and inhibition of apoptosis in the prostate suggests a potential to prevent and treat BPH or act as an indicator for post-therapy BPH status. *Desulfovibrio* is a Gram-negative sulfate-reducing bacterium. Recently, some *Desulfovibrio* spp. were reported to have bioremediation potential for toxic radionuclides, such as uranium, via a reductive bioaccumulation process [38]. In addition, *Desulfovibrio* was reported to interact with the surface of intestinal epithelial cells and induce apoptosis [39]. Similar to previous reports, the abundance of *Desulfovibrio* is related to apoptosis in the prostate. Therefore, *Desulfovibrio* has the potential to prevent and treat BPH or act as an indicator of post-therapy BPH status.

In conclusion, the decreased abundance of *Lactobacillus* and the increased abundances of *Flavonifractor*, *Acetatifactor*, *Oscillibacter*, *Pseudoflavonifractor*, *Butyricimonas*, and *Anaerotruncus* genera due to BPH induction are potential indicators for the diagnosis of BPH. *Lactobacillus*, *Anaerobacterium*, and *Desulfovibrio* genera, which were significantly increased by finasteride treatment, have potential as therapeutic agents for the prevention and treatment of BPH or as indicators of post-therapy BPH status. *Lactobacillus* and *Acetatifactor*, whose abundances were significantly altered by BPH induction and then significantly normalized by finasteride treatment, are the major gut microbes associated with the prostate profile, 5α-reductase, and apoptosis in the prostate during induction or amelioration of BPH. These results suggest that the altered abundance of specific gut microbes, caused by the induction of BPH and treatment with finasteride, are closely correlated with the regulation of prostate profiles, hormones, and apoptosis of the prostate, and these gut microbes could be used as both indicators and therapeutic measures.

## 4. Materials and Methods

### 4.1. Materials

Testosterone undecanoate was purchased from Bayer Co., Ltd. (Nebido^®^; Seoul, Republic of Korea) and finasteride was purchased from MSD (Proscar^®^; Seoul, Republic of Korea). Male Wistar Hannover rats were purchased from Samtako Co., Ltd. (Osan, Republic of Korea) and acclimatized for 1 week. The animals had ad libitum access to feed and water during the experiment. They were housed in a setting where the temperature was 20–24 °C and the relative humidity 30–70%, with a 12 h dark/light cycle. All experimental procedures were performed in compliance with the Sahmyook University test animal guidelines (SYUIACUC 2022-014).

### 4.2. BPH Induction and Finasteride Treatment Model

Male Wistar Hannover rats (300–370 g) were grouped (*n* = 7) into a control group, prostatic hyperplasia model group (BPH), and finasteride-treated group (BPH + Fina). The BPH and BPH+Fina groups were anesthetized using tiletamine/zolazepam (Zoletil 50; Virbac, France) and xylazine (Rompun; Bayer, Seoul, Republic of Korea). Castration was performed by removing the testicles, epididymis, and epididymal fat. The control group rats underwent a sham operation. Testosterone undecanoate (125 mg/kg) was injected subcutaneously every 3 weeks to induce BPH, and the control group rats were injected with corn oil instead [11]. Simultaneously, the BPH+Fina group rats were treated with 0.8 mg/kg of finasteride for 12 weeks.

### 4.3. Gut Microbiota Analysis

Gut microbiota analysis was performed using next-generation sequencing (NGS). Total DNA of the cecum, including that from the feces, was extracted using a Power Soil DNA Isolation Kit (Mo Bio Laboratories Inc.; Solana Beach, CA, USA). Isolated DNA was amplified using the 16S Amplicon PCR primer (forward: 5′-TCGTCGGCAGCGTCAGATGTGTATAAGAGACAGCCTACGGGNGGCWGCAG-3′, and reverse: 5′-GTCTCGTGGGCTCGGAGATGTGTATAAGAGACAGGACTACHVGGGTATCTAATCC-3′) for preprocessing. To attach the indices and Illumina sequencing adapters to the amplified PCR products, an index PCR was conducted using a Nextera^®^ XT Index Kit and Illumina sequencing adapters. After constructing the 16s rDNA V3 and V4 amplicon sequencing libraries, they were sequenced using an Illumina MiSeq instrument (2 × 300 paired-end sequencing). The adapter sequences were removed from the original paired-end reads using CutAdapt v1.11, and reads merged with the original paired-end reads were generated using FLASH v1.2.11. Reads with more than two ambiguous nucleotides or low-level reads shorter than 300 bp were filtered. Finally, potential chimeric reads were removed using UCHIME v4.2.40, based on the Bellerophon method.

Operational taxonomic units (OTUs) were calculated from the reads of each preprocessed sample. The OTU number was determined by clustering the sequences from each sample using a 97% sequence identity cut-off using QIIME software (v.1.8.0), and the alpha diversity was analyzed. Taxonomic abundance was calculated using RDP Classifier v1.1 from the reads of each pre-processed sample. Normalization of the microbial composition among samples was performed by dividing the taxonomic abundance count and the number of preprocessed reads for each sample. Principal component analysis (PCA) was performed after applying the Bray–Curtis distance through the identification of differences between organism compositions for beta diversity. Linear discriminant analysis (LDA, LEFSe) was used to estimate taxonomic abundances and identify differences between groups. MultiExperiment Viewer (MeV) software (v.4.8.1) was used to confirm the abundance of functional genes and to show the correlation using a heatmap.

### 4.4. Statistical Analysis

All results are expressed as mean ± standard deviation (SD), and the significance of differences was analyzed using a t-test followed by the Mann–Whitney U test as a post-hoc test. Relative abundance was analyzed using LEfSe based on the Kruskal–Wallis and Wilcoxon tests, and correlation analysis was performed using the Spearman correlation coefficient in GraphPad PRISM^®^ Version 5.0 (GraphPad Software; San Diego, CA, USA). Statistical significance was set at *p* < 0.05, *p* < 0.01, and *p* < 0.001.

## Figures and Tables

**Figure 1 ijms-24-05904-f001:**
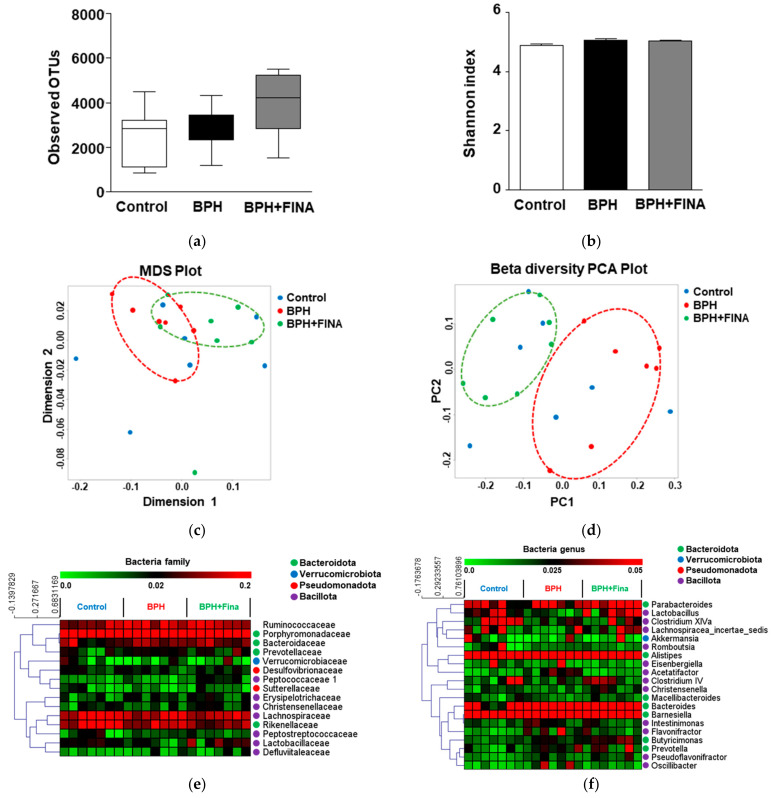
Microbial diversity in the bacterial community, categorized based on benign prostatic hyperplasia (BPH) and finasteride treatment. Observed operational taxonomic units (OTUs) (**a**), Shannon index (**b**), multidimensional scaling (MDS) plot (**c**), and principal coordinate analysis (PCoA) plot (**d**) of cecum samples of Wistar rats. Hierarchical clustering of most abundant bacterial families (**e**) and genera (**f**) are labeled in the heat map using Spearman’s rank correlation. The groups included: control, injected with corn oil following sham operation; BPH, injected with testosterone undecanoate (125 mg/kg) after castration; BPH+Fina, injected with testosterone undecanoate (125 mg/kg) and finasteride (0.8 mg/kg) after castration.

**Figure 2 ijms-24-05904-f002:**
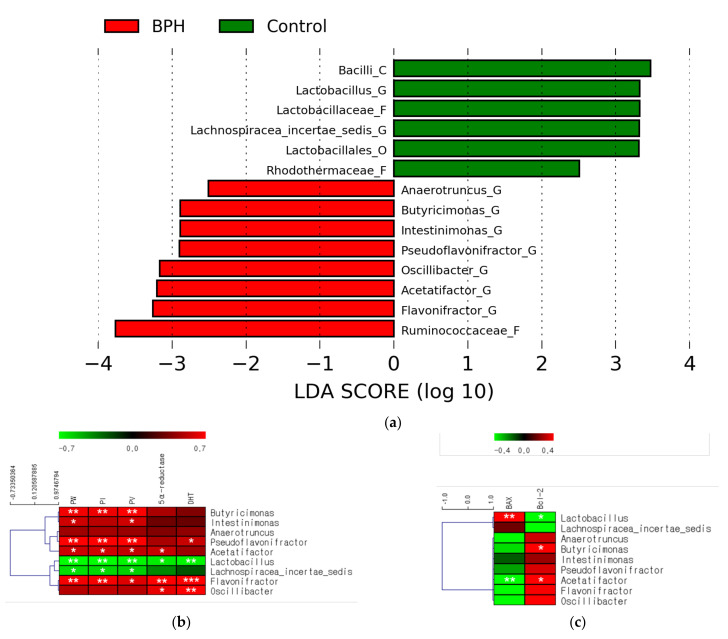
Alteration of relative bacterial abundance following BPH induction. Comparison between control and BPH groups (**a**). Significant differences (*p* < 0.05) were identified using LEfSe based on both the Kruskal−Wallis test (between classes) and the Wilcoxon test (between subclasses). The threshold logarithmic LDA score was 2.5. Correlation analyses between bacterial abundances and BPH indicators (**b**) or apoptosis markers (**c**) were conducted. Statistical significance (Spearman’s correlation coefficient): * *p* < 0.05, ** *p* < 0.01, and *** *p* < 0.001.

**Figure 3 ijms-24-05904-f003:**
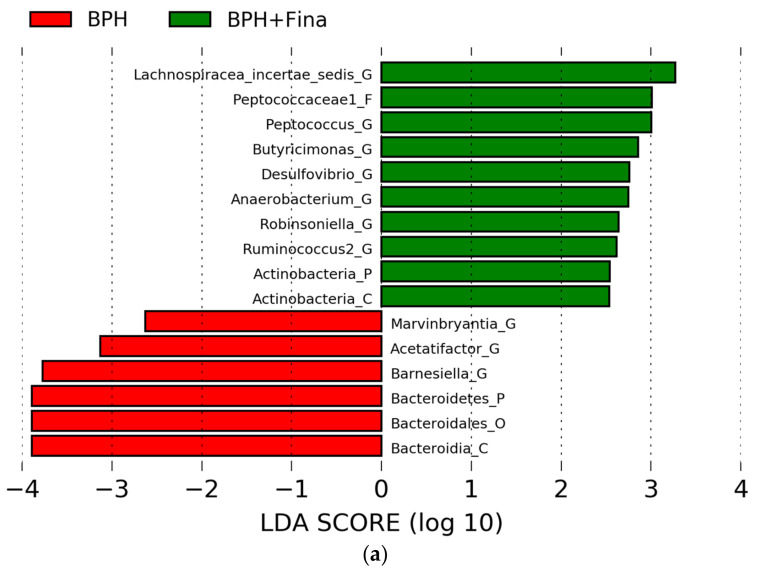
Alteration of relative bacterial abundances through finasteride treatment. Comparison between BPH and BPH+Fina groups (**a**). Significant differences (*p* < 0.05) were identified using LEfSe based on the Kruskal−Wallis test (between classes) and the Wilcoxon test (between subclasses). The threshold logarithmic LDA score was 2.5. Correlation analyses between bacterial abundances and BPH indicators (**b**) or apoptosis markers (**c**) were conducted. Statistical significance (Spearman’s correlation coefficient): * *p* < 0.05, ** *p* < 0.01, and *** *p* < 0.001.

**Figure 4 ijms-24-05904-f004:**
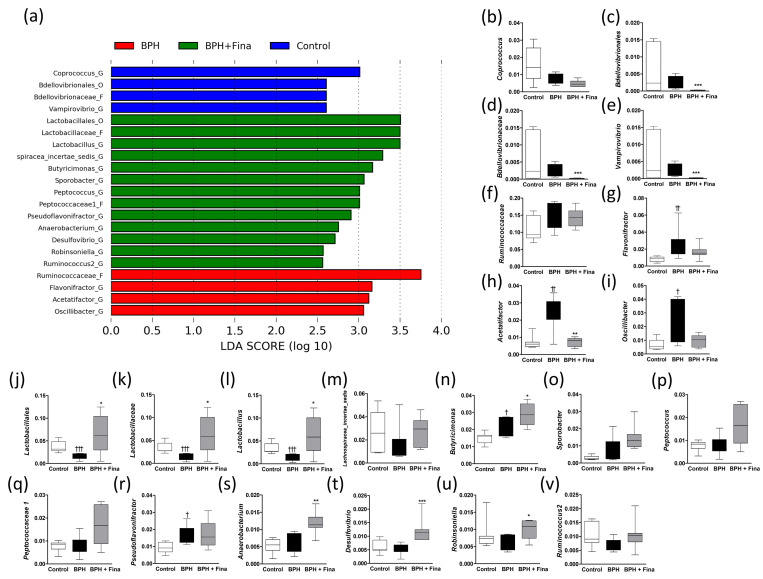
Alteration of relative bacterial abundance due to BPH induction and finasteride treatment. Comparison between control, BPH, and BPH+Fina groups (**a**). The microbial abundance based on LEfSe analysis is shown in (**b**–**v**). Significant differences were identified using LEfSe (*p* < 0.05) based on both the Kruskal–Wallis test (among classes) and Wilcoxon test (between subclasses). The threshold logarithmic LDA score was 2.5. Statistical analyses were performed using t-test and Mann–Whitney test. ^†^
*p* < 0.05, ^††^
*p* < 0.01, and ^†††^
*p* < 0.001 compared with control group; * *p* < 0.05, ** *p* < 0.01, *** *p* < 0.001 compared with the BPH group. The groups included: control, injected with corn oil following sham operation; BPH, injected with testosterone undecanoate (125 mg/kg) after castration; BPH+Fina, injected with testosterone undecanoate (125 mg/kg) and finasteride (0.8 mg/kg) after castration.

## Data Availability

Data is contained within the article or Appendix A.

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
