# Peer review of "Alteration of Gut Microbes in Benign Prostatic Hyperplasia Model and Finasteride Treatment Model"

_ijms, 2023, doi:10.3390/ijms24065904_

Round 1

Reviewer 1 Report (Previous Reviewer 2)

No additional comments. This is a nice study identifying and investigating the microbiota-prostate axis.

Reviewer 2 Report (Previous Reviewer 1)

All the previous comments were addressed ,thank you

This manuscript is a resubmission of an earlier submission. The following is a list of the peer review reports and author responses from that submission.

Round 1

Reviewer 1 Report

This study evaluated the alteration of gut microbes in benign prostatic hyperplasia. The organization is good and the topic is interesting. This manuscript needs minor revision before publication

1-In the abstract, it was mentioned that increased acatatifactor was associated with inhibition of apoptosis then the opposite was stated in lines 21 and 22  please clarify

2-figure 1,3and 5 are not clear, please improve the resolution

3- gut microbiota analysis in materials and methods needs to be explained in detail

Reviewer 2 Report

An et al explore the influence of a high-DHT / BPH inducing model, and BPH plus finasteride treatment on the gut microbiota in male rats. Significant shifts in the micorbiota of the BPH group occurred, notably in the abundance of Lactobacillus and Acetatifactor, which were normalized by finasteride treatment. This work identifies potential microbiota compositional markers for BPH, as well as probiotic treatments. Following are the reviewer comments:

Figure S2 shows nice data prostate data that supports the BPH in vivo model. Consider referencing the figure as validation of the model early in the result section, or even presenting it as Figure 1 in the main text.

In regard to the conclusions associated with Figures 3 and 5: although correlations between the composition of the gut microbiota and BPH indicators show significance, many of the indicators did not show significant differences in the prostate across all groups, such as PSA and all inflammatory markers. This brings into question the significance of some of the correlation analyses. If the correlation was physiologically relevant one would expect corroborating results in the prostate tissue. At minimum the correlations showing significance without corroborating prostate tissue data need to be explicitly acknowledged in the text as such, and the reader referenced to the supplementary data. The reviewer recommends only presenting and drawing conclusions from correlations that have corroborating prostate tissue data, and present the associating prostate tissue data in the main text as well.

The text descriptions for Figures 3 and 5 are identical. This is unacceptable, please add more information detailing the experiments the graphs describe.

Figures 2, 4, and 6 are redundant, and can be streamlined. In the current state, each taxon is a separate graph - this data would be more easily observed by the reader if was condensed into one or two graphs, showing multiple taxa on each graph. The redundancy is in that the control vs BPH, then BPH vs BPH+fina, then control vs BPH vs BPH+fina are all presented separately. It is recommended that the manuscript be streamlined so that only the control vs BPH vs BPH+fina data is presented.

Some introduction into finasteride as a treatment for BPH through 5a reductase inhibition in the introduction may be beneficial to some readers.

Some graphs have text that is difficult to read, all text needs to be clearly legible. 

Round 2

Reviewer 2 Report

Although the addition of section 2.1 is a welcome addition to the manuscript, it does not sufficiently address Point 2. The manuscript presents and makes conclusions heavily reliant upon correlations between changes in the gut microbiota and BPH markers that do not show significant differences in the prostate tissue. For example, the last sentence of the discussion, arguably the main conclusion of the manuscript, states "These results suggest that the altered abundance of specific gut microbes, caused by the induction of BPH and treatment with finasteride, are closely correlated with the regulation of BPH indicators, apoptosis of prostate, and inflammatory cytokines, and these gut microbes could be used as both indicators and therapeutic measures." As there were no significant differences in the prostate inflammatory cytokines across the experimental groups (Figure S4), how are the microbiota-prostate marker correlations, although themselves showing significance, physiologically relevant and valid? The prostate markers that showed a significant change in the BPH groups and corresponding significant reversal with finasteride treatment are 5-alpha reductase, DHT, Bax, and Bcl-2. The other prostate markers analyzed do not show significance across the groups, invalidating the corresponding microbiota-prostate marker correlations.